# Logic-Aware Knowledge Graph Reasoning for Structural Sparsity under Large Language Model Supervision

## ABSTRACT

Knowledge Graph (KG) reasoning aims to predict missing entities in incomplete triples, which requires adequate structural information to derive accurate embeddings. However, KGs in the real world are not as dense as the idealized benchmarks, where sparse graph structures restrict the comprehensive structural information for superior performance. Although the logical semantics in KGs shows its potential in alleviating the impact of structural sparsity, there still exist some challenges. The deficient supervision and the semantic gap of logic make it difficult to introduce logical semantics in sparse KG reasoning. To this end, we propose a novel KG reasoning approach LoLLM[1] injecting logic with the supervised information supplied by the Large Language Model (LLM), which is proved to be effective in evaluating and scoring. Firstly, LoLLM derives structural embeddings employing a graph convolutional network (GCN) with relation-aware and triple-aware attention. LoLLM secondly constructs reasoning paths instantiated from the first-order logics extracted from sparse KGs, and injects the logical semantics by a designed LLM-enhanced tuning strategy. We propose a textual loss (TL) and a logical loss (LL) in the optimization and obtain logical tuning embeddings of KG in this process. Finally, LoLLM fuses structural embeddings from the GCN and logical tuning embeddings from the LLM-enhanced tuning for scoring and incomplete triple prediction. Extensive experiments on two sparse KGs and a benchmark show that LoLLM outperforms state-of-the-art structure-based and Language Model (LM)-augmented baselines. Moreover, the logics with corresponding confidences provide explicit explanations as an interpretable paradigm.

## CCS CONCEPTS

• **Computing methodologies → Knowledge representation and reasoning**.

## KEYWORDS

Sparse Knowledge Graph Reasoning, Knowledge Graph, First-order Logic, LLM-enhanced Tuning

---

[1]KGs provide a data and knowledge management on the Web, and the KG embedding and reasoning are in the scope of Semantics and Knowledge Track.

---

**ACM Reference Format:**
Anonymous Author(s). 2018. Logic-Aware Knowledge Graph Reasoning for Structural Sparsity under Large Language Model Supervision. In . ACM, New York, NY, USA, 12 pages. https://doi.org/XXXXXXX.XXXXXXX

## 1 INTRODUCTION

Knowledge graphs (KGs) possess massive triples consisting of entities and relations in a structured storage scenario, representing real-world knowledge. They have benefited in various downstream tasks, such as question answering [1, 7], generation task [9], and information extraction [10], etc. According to the open-world assumption, the KGs are not complete, so it is essential to predict the missing entities or relations in incomplete triples. Some dominant methods [28, 36] assume the abundant structural information in the prediction. They obtain relation and entity embeddings on widely used benchmarks [2] with sufficient structure.

Some KGs, unlike benchmarks, have sparse connections between nodes [17]. For instance, the density (calculated as $\frac{\#T}{\#E(\#E-1)}$) of CN-100K [30] is about $1/100$ compared to the general benchmark FB15K-237 [2], which is shown in Table 1. This situation, which is regarded as the **structural sparsity**, will worsen the embedding of entities and relations, thereby affecting the reasoning performance. Aiming at the structural sparsity, some methods [19, 37] concatenate textual embeddings of entities to improve KG embeddings, whereas they have difficulties in distinguishing massive entities in sparse KGs. For example, in Fig. 1, the structure of the KG is not able to provide comprehensive semantics of entity *Liquid* and *Fluid* and distinguish them in reasoning. Even with both the structure and entity text "Liquid" and "Fluid", the embeddings (denoted as $A$ and $B$ in Fig. 1) are still not distinct enough to correctly predict the query $(Water, \text{hasProperty}, ?)$. Focusing on this, some other methods [13, 41] leverage densification strategy by adding triples with the embedding similarity to improve the sparse structure, which is still strongly related with the precise embeddings.

In recent studies, the first-order logic rules [20, 38] are introduced in KGs to benefit reasoning. As shown in Fig. 1, the query $(Water, \text{hasProperty}, ?)$ can be correctly predicted with the semantics of the following first-order logic:

$$\alpha \underbrace{\text{isA}(X,Z) \wedge \text{hasProperty}(Z,Y)}_{body} \rightarrow \underbrace{\text{hasProperty}(X,Y)}_{head}, \quad (1)$$

which consists of atoms, i.e. $\text{hasProperty}(X, Y)$, connected by a conjunction $\wedge$. Given the semantics of logic and the instantiated triples $(\{Liquid, \text{isA}, Water\}, \{Water, \text{hasProperty}, Fluid\})$, the embeddings of two entities can be corrected in a scenario of structural sparsity and the reasoning result $(Water, \text{hasProperty}, Fluid)$ will be correctly derived. However, even with the first-order logic, there still exist some important issues to be solved:

**(1) Deficient Supervision.** When injecting the semantics of the logic, we should not only pay attention to the body, i.e.

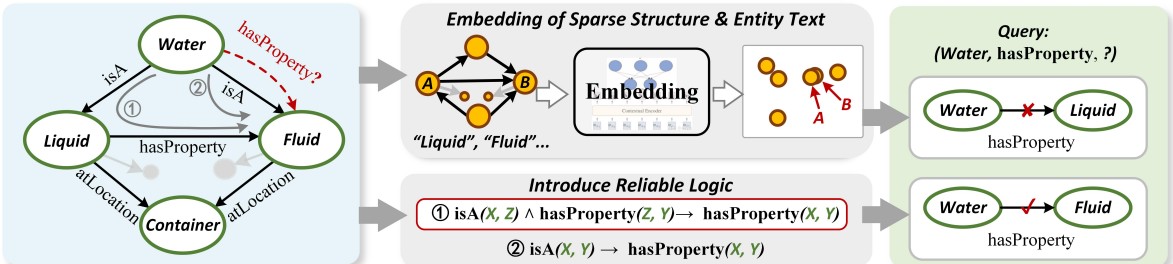

**Figure 1: Reliable logic helps get a correct KG reasoning result under the undistinguished embeddings with structural sparsity.**

**Table 1: Statistics of the datasets.**

| Dataset | #E | #R | #T | Density |
|---|---|---|---|---|
| CN-100K | 78,088 | 34 | 100,000 | $1.64e^{-5}$ |
| FB15K-237-Sparse | 14,541 | 237 | 18,506 | $8.72e^{-5}$ |
| FB15K-237 | 14,541 | 237 | 272,115 | $1.28e^{-3}$ |

$isA(X, Z) \wedge hasProperty(Z, Y)$, but also capture whether the logic is reliable in reasoning, which is denoted as the confidence $\alpha$ in Eq. (1) of the logic [45]. For example, in Fig. 1, the logic ① is more reliable than ② during predicting $(Water, hasProperty, Fluid)$, then the confidence of ① will be greater than that of ②. Previous logic learning methods cannot provide the supervised confidences of first-order logics, while they employ the attention weight to evaluate the logic instead [38]. These methods have challenges to get the supervised confidences under the scenario of KG structural sparsity, because the attention weights are still calculated by the KG embeddings. **(2) Semantic Gap.** As shown in Fig. 1, the first-order logic is constructed by symbols (i.e. atoms, $\wedge$, $\rightarrow$), which is discrete and difficult to be fused with KG embeddings or text embeddings. This phenomenon is regarded as a semantic gap between the logic and continuous KG embedding space [43]. Specifically, we intend to introduce the conjunction ($\wedge$) semantics of the body, i.e. $isA(X, Z) \wedge hasProperty(Z, Y)$, into the embedding model when predicting $(Water, hasProperty, ?)$. Besides, $X, Y, Z$ are variables in the first-order logic, which are instantiated as various entities in different reasoning process. These pieces of information represented as discrete symbols in logics are necessary to be captured by continuous embedding models.

To address these issues, we propose a **Lo**gic-aware knowledge graph reasoning method solving structural sparsity under Large Language Model (**LLM**) supervision, named as LoLLM. In LoLLM, we innovatively leverage LLM to provide supervised information for the logic, specifically the confidences, inspired by the abundant background of LLM [12], which is good at scaling and evaluating items. In order to obtain accurate embeddings in the sparse KGs, we propose a logical tuning embedding module. In detail, LoLLM firstly derives structural embeddings employing a graph convolutional network (GCN), which contains relation-aware and triple-aware attention. LoLLM secondly instantiates first-order logics as reasoning paths extracted from sparse KGs, and injects the logical semantics by a designed LLM-enhanced tuning process. In this process, the LLM is used for scaling the confidence of first-order logic by reasoning paths, aiming at the deficient supervision of structural sparsity. Then, considering the semantic gap between first-order logic and the embedding space, we propose a textual loss (TL) and a

logical loss (LL) in fine-tuning pretrained language models (PLMs) to introduce semantics of reasoning paths and obtain logical tuning embeddings. Finally, LoLLM fuses structural embeddings from the GCN and logical tuning embeddings from the LLM-enhanced tuning process to score the triple and implement incomplete triple prediction. Meanwhile, the logics with confidences representing the reliability of first-order logic in reasoning.

Our main contributions are in the following three folds:

- A novel logic-aware KG reasoning method for structural sparsity by LLM is proposed. To the best of our knowledge, it is the first method to model and combine logical tuning embeddings with the LLM for accurate KG embeddings.
- In order to solve the deficient supervision in introducing logics, we use the LLM to provide supervised information of logics in the LLM-enhanced tuning process. As for the semantic gap between discrete logic and embedding models, we instantiate the logic as reasoning paths and design two novel losses (TL and LL) in the optimization.
- Extensive experiments on two sparse KG and a benchmark KG show that LoLLM achieves outstanding effectiveness compared to SOTA baselines. Meanwhile, the logics with corresponding confidences provide reliable explicit explanations for the reasoning process.

## 2 RELATED WORK

### 2.1 Knowledge Graph Reasoning

KG reasoning is to predict missing entities and relations of the incomplete triples by their embeddings. Based on the depending information, we divide the methods into two categories.

**Structure-based** methods mainly focus on structural information for the embeddings. Translation-based methods translate the entities and relations into a low-dimensional continuous vector space, which include TransE [3], TransH [40], etc. Semantic factorization-based methods utilize the semantic factorization to obtain representations, which usually represent relations as matrices, such as RESCAL [22] and DisMult [44]. RotatE [32] defines each relation as a transformation matrix from the source entity to the target entity in the complex vector space. With the development of deep learning, neural networks are used to capture structural information. ConvE [5] and ConvTransE [29] obtain entity and relation embeddings by convolutional networks. Recent methods model the graph structure of KGs. R-GCN [28] and CompGCN [35] capture the neighborhood information and model the relational data by a GCN.

 

**LM-augmented** methods frame KG embeddings via language models, which have led a significant improvement [24, 48]. More studies leverage fine-tuning PLMs for the KG embeddings. KG-Bert [46] obtains embeddings considering the triple as a sequence and link prediction as a sequence classification task. KEPLER [39] encodes entity descriptions with a PLM as their embeddings, and simultaneously optimizes the knowledge embeddings and language modeling objectives. StAR [36] proposes a structure-augmented text representation by PLMs to improve KG embeddings. In addition, some studies employ prompt tuning for KG embeddings and reasoning. For example, PKGC [18] transfers triples into natural prompt sentences for KG embeddings. Moreover, LLMs are being explored for their potential in KG reasoning [49], in which a zero-shot reasoning scenario is created to simulate the KG reasoning.

Previous KG reasoning methods are good at the idealized benchmarks with adequate structural information. Although some methods focus on the sparse KGs, but they still strongly depend on the embeddings [17, 19]. Distinguished with them, we inject the logical semantics for the structural sparsity, which is crucial to enhance the effectiveness of sparse KG reasoning.

## 2.2 Logical Reasoning

Logical reasoning focuses on introducing the semantic of logic rules into reasoning process, which can not only improve the reasoning performance but also give an interpretable reasoning process. Logical reasoning can be implemented in various areas. For the visual reasoning task, NSVQASP [8] uses a neuro-symbolic visual question answering (VQA) architecture which disentangles perception from reasoning provided by logical theory. For the text reasoning tasks, Logiformer [43] utilizes a two-branch graph transformer network for logical question answering. It constructs implicit logical units from the text and improves the performance in logical reasoning task. As for the graph networks, SGR [16] implements reasoning through a group of symbolic nodes whose outputs explicitly represent different properties of semantics in a prior graph.

Inspired by the key technique of previous logical reasoning in text, images and graphs, we innovatively introduce the logical reasoning into KG reasoning, especially the reasoning on sparse KGs for comprehensive KG embeddings.

## 3 PRELIMINARY

### 3.1 KG Embedding and Reasoning

A KG can be denoted as $G = \{R, E, T\}$, in which $R, E$ are the sets of relations and entities, respectively. $T \subset E \times R \times E$ is the set of triples representing facts or commonsenses. A target triple is denoted as $(s, r, o)$, where $s, o \in E$ are subject and object respectively, and $r \in R$ refers to the predicate in the triple. KG reasoning aims at predicting the missing entity in the incomplete triple $(s, r, ?)$ or $(?, r, o)$. The prediction is maximizing the score of triple $(s, r, o)$ based on the low-dimensional vector embeddings of entities and relations in $G$.

### 3.2 First-Order Logic

In KGs, the implicit generalized logic for reasoning is the first-order logic [20, 38], which is denoted as a Horn clause [26]:

$$\alpha \; \exists x, z_1, \cdots, z_{n-1}, y, \mathsf{r}_1(X, Z_1) \wedge \cdots \wedge \mathsf{r}_n(Z_{n-1}, Y) \rightarrow \mathsf{r}(X, Y), \quad (2)$$

**Table 2: Important symbols and their descriptions.**

| Symbol | Description |
|---|---|
| $(s, r, o)$ | Target triple in the sparse KG. |
| $v_i^{(l)}$ | The structural embedding of entity $i$ at layer $l$. |
| $\mathcal{P}_{s \rightarrow o}$ | Reasoning paths in $G$ from $s$ to $o$. |
| $L_m$ | Maximum length of the reasoning path. |
| $p_k$ | The $k$-th reasoning path in $\mathcal{P}_{s \rightarrow o}$. |
| $\tau_k$ | The reasoning sentence corresponding to $p_k$. |
| $\mathcal{T}(s, r, o)$ | The reasoning paragraph around $(s, r, o)$. |
| $\langle i \rangle$ | The text of the entity $i$. |
| $z_{\langle i \rangle, k}$ | The text embedding of $k$-th $\langle i \rangle$. |
| $z_i$ | The mean textual embedding of entity $i$. |
| $w_{k,o}$ | The similarities of $o$ between in $\tau_k$ and $\mathcal{T}(s, r, o)$ generated by LLM. |
| $\overline{w}_{k,o}$ | The confidence of $\tau_k$ in $\mathcal{T}(s, r, o)$ and the corresponding first-order. |
| $h_{\langle i \rangle}$ | The logical tuning embedding of the entity $i$. |

in which the body and head consist of atoms, e.g. $\mathsf{r}(X, Y)$. The atoms in body are combined by the conjunction symbol $\wedge$ and point to the head atom by an implication symbol $\rightarrow$. Each atom is constructed by a predicate (e.g. $\mathsf{r}$), which refers to the relation in KGs, and two arguments (e.g. $X, Y$), which can be instantiated as entities. The adjacent atoms in the body share the same argument, since the first-order logic aims to construct a closed path in KGs.

In addition, $\alpha$ denotes the confidence of the first-order logic, which evaluates the reliability of the logic rule during reasoning. In the reasoning scenario, instantiated logic rules have various values. The implicit first-order with confidence provides interpretability of the reasoning model.

### 3.3 Reasoning Path

In the KG reasoning scenario, first-order logics are instantiated as the reasoning paths with their own confidence. The arguments $X, Z_1, \cdots, Z_{n-1}, Y$ in rule (2) are instantiated as entities $x, z_1, \cdots, z_{n-1}, y$ in a reasoning path, which is denoted as $x \xrightarrow{\mathsf{r}_1} z_1 \cdots z_{n-1} \xrightarrow{\mathsf{r}_n} y$. For instance, a reasoning path $Liquid \xrightarrow{\text{AtLocation}} Cup \xrightarrow{\text{IsA}} Container$ in Fig. 2 is instantiated from the body atoms of the following first-order logic:

$$\text{AtLocation}(X, Z) \wedge \text{IsA}(Z, Y) \rightarrow \text{AtLocation}(X, Y). \quad (3)$$

The reasoning path is constructed by two connected triples ($\{Liquid, \text{AtLocation}, Cup\}$, $\{Cup, \text{IsA}, Container\}$). Each triple is an instantiated *proposition* [14], which is a significant component in first-order logic.

## 4 METHODOLOGY

In this section, we demonstrate LoLLM with the help of Fig. 2. Specifically, LoLLM is in three parts. The main contents are in the first two part, which refer to the structural and logical branches in the framework for KG reasoning.

### 4.1 Structural Modeling

As mentioned in Section 3.1, LoLLM aims at scoring the target triple $(s, r, o)$ in $G$. Considering the plenty of structural information in the

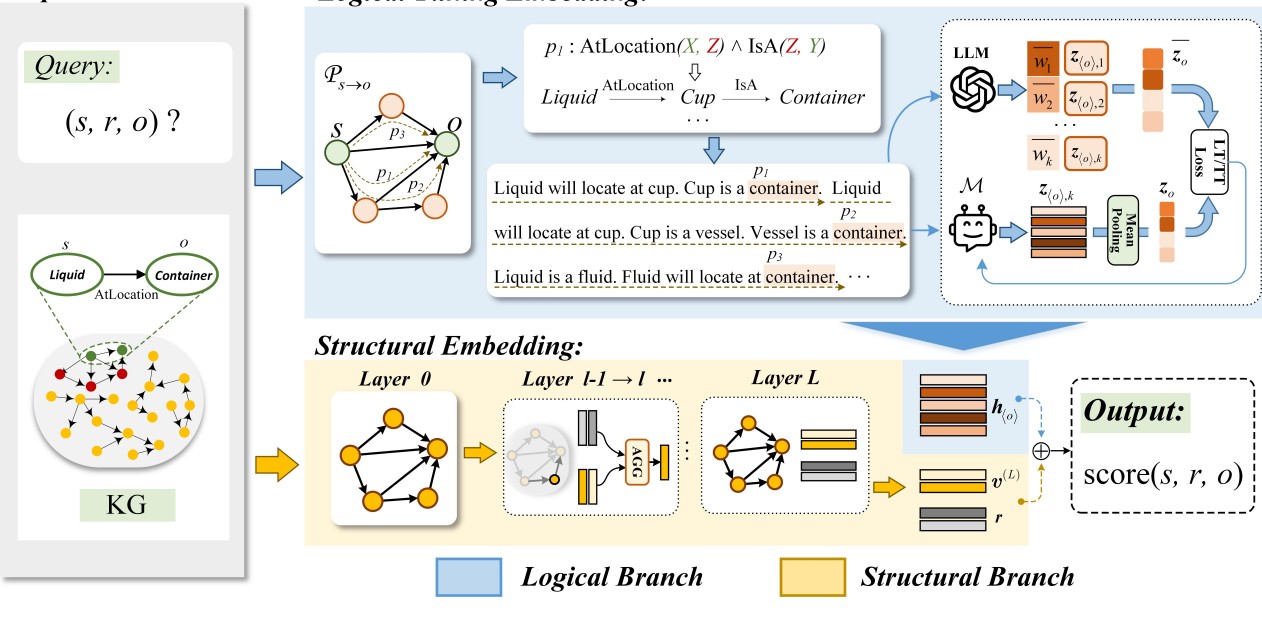

**Figure 2: The overall framework of LoLLM, which consists of a structural branch and a logical branch. LoLLM firstly obtain structural KG embeddings by a GCN. Then, it obtains the logical embedding by a logical tuning embedding tuning, which contains LLM-enhanced tuning process for the deficient supervision. Finally, LoLLM combine the embeddings from two branches and obtain the score of the target triple.**

KG, we employ the graph convolutional network (GCN) [19, 28] for structural modeling. In particular, we use an $L$-layer GCN for the relation-aware and triple-aware attention based on the structure to obtain entity and relation embeddings. The forward-pass update from $l$-th layer to $l + 1$-th layer is defined as:

$$v_i^{(l+1)} = \phi_1\left(\sum_{r \in R, j \in \mathcal{N}_r(i)} \theta_{(i,r,j)}^{(l)} \mathbf{W}_r^{(l)} v_j^{(l)} + \theta_0 \mathbf{W}_0^{(l)} v_i^{(l)}\right), \quad (4)$$

where $v_i^{(l+1)}$ refers to the embedding of entity $i$ at $l + 1$-th layer. $i, j$ indicate a pair of entities connected by relation $r$. $\mathcal{N}_r(i)$ refers to the set of neighbor entities of $i$ connected by the relation $r$. $\mathbf{W}_r^{(l)}$ and $\mathbf{W}_0^{(l)}$ refer to the transformation matrices during the forward-pass update. $\theta_0$ decides whether the self-attention is employed. $\phi_1(\cdot)$ indicates the activation function, specifically $\tanh(\cdot)$ in Eq. (4). To evaluate the neighbor attention of entity $i$, we calculate $\theta_{(i,r,j)}$ as the triple-aware attention weight of the triple $(i, r, j)$:

$$\theta_{(i,r,j)}^{(l)} = \theta_r \mathbf{W}(v_i^{(l)} \oplus v_j^{(l)}), \quad (5)$$

in which $\mathbf{W}$ is the transformation matrix, and $\theta_r$ refers to the specific relation attention-aware weight connecting entity $i$ and $j$. $\oplus$ concatenates two vector embeddings. The structural modeling process generates embeddings of entities and relations, which will be significant in scoring the target triple.

## 4.2 Logical Tuning Embedding

As for the reasoning paths instantiated from first-order logics, we design a strategy to introduce them into the encoding language model to obtain embeddings of entities and relations. This is the main part of LoLLM, and we propose an LLM-supervised method to implement the logical tuning for precise KG embedding.

*4.2.1 Logic Extraction and Injection.* In this process, we construct the instantiated logic rules as the reasoning paths, which has been illustrated in Section 3.3. As previously discussed, the reasoning paths contain critical semantics of the first-order logics during KG reasoning, so we intend to introduce the critical semantics of the reasoning path when embedding entities and relations. Given the target triple $(s, r, o)$, we firstly extract the paths consist of connected triples between head $s$ and tail $o$ by breadth first search (BFS) algorithm [4]. Considering the time and space, the length of the reasoning paths is constrained in a threshold $L_m$. The set of the reasoning paths from $s$ to $o$ is denoted as $\mathcal{P}_{(s,o)} = (p_1, p_2, \cdots, p_k, \cdots, p_K)$. Each reasoning path $p_k : x \xrightarrow{r_1} z_1 \cdots z_{n-1} \xrightarrow{r_n} y$ is instantiated from the first-order logic rule $l_k : r_1(X, Z_1) \wedge r_2(Z_1, Z_2) \cdots r_n(Z_{n-1}, Y)$. For example, in Fig. 2, there are 3 reasoning paths $p_1, p_2, p_3$ when $L_m$ is preset as 3, in which $p_1 = (Liquid, \text{AtLocation}, Cup, \text{IsA}, Container)$.

In order to introduce the discrete logic rule $l_k$ and reasoning path $p_k$ into the continuous model, we propose a logic injection strategy. Given the set of reasoning paths $\mathcal{P}_{(s,o)}$, we construct the reasoning sentence based on the template we design, which is denoted as $\tau(\cdot)$. After transformation, the reasoning paragraph $\mathcal{T}(s, r, o)$ around the target triple $(s, r, o)$ is denoted as following:

$$\tau_k = \overbrace{\tau(s_1, r_1, o_1) \oplus \tau(s_2, r_2, o_2) \cdots \oplus \tau(s_n, r_n, o_n)}^{p_k}, \quad (6)$$

$$\mathcal{T}(s, r, o) = [\tau_1 \oplus \tau_2 \cdots \oplus \tau_k \cdots \oplus \tau_K] \oplus q \oplus o, \quad (7)$$

**Figure 3: The generation of similarities by the LLM.**

in which $\tau(\cdot)$ constructs the textual sentence of a triple. For example, $\tau(Liquid, AtLocation, Cup)$ is *"Liquid will locate at cup."* $\oplus$ is the concatenation operation of sentences to simulate the $\wedge$ semantics. $q$ is the question transferred from the relation $r$ and subject $s$.

*4.2.2 LLM-Enhanced Tuning.* As a significant part of deriving logical semantics from the reasoning paragraph, we propose an LLM-enhanced tuning strategy. Considering the lacking supervision, we introduce the LLM to provide supervised information. In Fig. 2, LLM and PLM are simultaneously applied to obtain precise embeddings of the entities. Inspired by the LLM-guided approaches [31, 47], we query LLM with prompt rather than fine-tuning the whole LLM.

With a PLM $\mathcal{M}$, we can obtain the embedding of object text $\langle o \rangle$ in $(s, r, o)$ from $\mathcal{T}(s, r, o)$. In this process, we aggregate the hidden states of multiple $\langle o \rangle$ tokens in $\mathcal{T}(s, r, o)$ and leverage the mean pooling to derive the embedding $z_o$ of $o$:

$$z_o = \sum_{k=1}^{|\langle o \rangle|} z_{\langle o \rangle, k}, \tag{8}$$

$$z_{\langle o \rangle, k} = \mathcal{M}(\langle o \rangle_k | \mathcal{T}(s, r, o)), \tag{9}$$

in which $|\cdot|$ gives the number of appearences of $\langle o \rangle$ in the $\mathcal{T}(s, r, o)$. $\langle \cdot \rangle$ generates the text of the entity. Alternatively, we can also focus on the embedding $z_s$ of $z$, which can be derived through a similar process: $z_{\langle s \rangle, k} = \mathcal{M}(\langle s \rangle_k | \mathcal{T}(s, r, o))$, and get $z_s$ by the mean pooling as well.

In order to generate the supervised samples, we employ the LLM for the weights of each reasoning path $p_k$ in the reasoning paragraph $\mathcal{T}(s, r, o)$, which obtain superior performance in scoring and evaluation [50]. As shown in Fig. 3, with the given prompt, the LLM generates the similarity of $\langle o \rangle$ between that in $\tau(s, r, o)$ and in each reasoning sentence in $\mathcal{T}(s, r, o)$, which is denoted as $w_{k,o}$. As the LLM will generate the similarity $w_{k,o} > 1$, we calculate the softmax to constrain the weight $\overline{w}_{k,o} \in [0, 1]$ and derive the LLM-based embedding of $\langle o \rangle$ by aggregating the embeddings:

$$w_{k,o} = \text{LLM}(\mathcal{T}(s, r, o)), \tag{10}$$

$$\overline{w}_{k,o} = \text{softmax}(w_{k,o}) = \frac{\exp(w_{k,o})}{\sum_{k'=1}^{|\langle o \rangle|} \exp(w_{k',o})}, \tag{11}$$

$$\overline{z}_o = \overline{w}_{k,o} \cdot z_{\langle o \rangle, k}, \tag{12}$$

Alternatively, we can employ the normalization to achieve this as well. The generated $\overline{w}_{k,o}$ is considered to be the confidence of logic, supplying the lacking supervised information during reasoning. The complete prompt is in Appendix C.3.

As for introducing the logical information, we fine-tune $\mathcal{M}$ with the **mask predicting task** using the following textual loss (TL):

$$\mathcal{L}_{TL} = -\frac{1}{N} \sum_{u=1}^{N} \log P(X_m^u | \mathcal{T}(s, r, o); \mathcal{M}), \tag{13}$$

$$= -\frac{1}{N} \sum_{u=1}^{N} \log \frac{\exp(z_m \cdot \mathbf{W}_m^u)}{\sum_{j \in \mathcal{V}} \exp(z_m \cdot \mathbf{W}_j)}, \tag{14}$$

where $N$ is the number of masked tokens, and $X_m^u$ refers to the $u$-th masked token in the $\mathcal{T}(s, r, o)$. $\mathbf{W}_m^u$ and $\mathbf{W}_j$ are weights corresponding to $u$-th masked token and all the candidate tokens in the corpus $\mathcal{V}$, respectively. $z_m$ represents the hidden embedding of the masked token. Moreover, as we discussed before, we try to reduce the effectiveness of different entities in the reasoning paths in order to introduce the generalization of first-order logics. During the process, we choose the margin-based loss [40] to focus on the overall object embedding and infoNCE loss [34] for every single appearance of object $o$ compared with the masked $o$. Therefore, we design a logical loss (LL) as following:

$$\mathcal{L}_{LL} = \overbrace{\max(0, D(z_o, \overline{z}_o) - D(z_{o^-}, \overline{z}_o) + \gamma)}^{\text{overall } o \text{ embedding}}$$
$$- \underbrace{\log \frac{\exp(\overline{z}_o \cdot z_{o,m}/\mu)}{\sum_i \exp(z_{o,i^-} \cdot z_{o,m}/\mu) + \exp(\overline{z}_o \cdot z_{o,m}/\mu)}}_{\text{masked } o \text{ embedding}}, \tag{15}$$

in which $D(\cdot)$ calculates the distance between two vectors. $z_{o^-}$ and $z_{o,i^-}$ refer to the negative samples, specifically other tokens instead of $\langle o \rangle$ in $\mathcal{T}(s, r, o)$. In Eq. (7), we replace the last $o$ with a masked token, which is denoted as $z_{o,m}$. $\gamma$ is a hyper-parameter in the margin-based loss and $\mu$ is the temperature in infoNCE loss, whose default is 1. In B-LoLLM, we add the margin-based loss and infoNCE loss to LL focusing on the subject $s$ with the same process as Eq. (15). We introduce the details of B-LoLLM in Appendix A. The overall loss is designed as:

$$\mathcal{L} = \lambda \mathcal{L}_{LL} + (1 - \lambda) \mathcal{L}_{TL}. \tag{16}$$

By fine-tuning the PLM, LoLLM can obtain precise embeddings of the entities of $G$ by comprehensive logical semantics for better initialized representations in triple scoring.

## 4.3 Scoring and Prediction

After fine-tuning the PLM $\mathcal{M}$ with the triples and reasoning paths, we input $\langle o \rangle$ (or $\langle s \rangle$) and take the hidden states as the logical embeddings of the entity:

$$h_{\langle o \rangle} = \mathcal{M}(\langle o \rangle). \tag{17}$$

To fuse the logical tuning embeddings and structural embeddings for prediction, we concatenate the embeddings obtained from the previous sections:

$$\psi_i = v_i^{(L)} \oplus h_{\langle i \rangle}, \tag{18}$$

**Table 3: Comparison results (%) of KG reasoning task on CN-100K, FB15K237-Sparse and FB15-237. ♣ indicates the results are from [13]. ◇ means the results are from [17]. The optimal and suboptimal values are marked in bold and underline respectively. The reasoning results on FB15K-237 are from [17] and [36].**

| Model | CN-100K | | | | | FB15K237-Sparse | | | | | FB15K237 | | | | |
| --- | --- | --- | --- | --- | --- | --- | --- | --- | --- | --- | --- | --- | --- | --- | --- |
| | H@1 | H@3 | H@10 | MRR | MR | H@1 | H@3 | H@10 | MRR | MR | H@1 | H@3 | H@10 | MRR | MR |
| *Structure-based* | | | | | | | | | | | | | | | |
| DisMult [44] ◇ | 4.51 | 9.76 | 17.44 | 8.97 | - | 9.20 | 14.60 | 22.30 | 13.60 | 3061 | 19.90 | 30.10 | 44.60 | 28.10 | 512 |
| ComplEx [33] ◇ | 7.42 | 12.45 | 19.01 | 11.40 | - | 9.10 | 14.30 | 21.60 | 13.20 | 3333 | 19.40 | 29.70 | 45.00 | 27.80 | 546 |
| ConvE [5] ◇ | 13.97 | 22.91 | 34.02 | 20.88 | - | 10.60 | 16.50 | 25.80 | 15.60 | 2263 | 22.50 | 34.10 | 49.70 | 31.20 | 245 |
| ConvTransE [29] ◇ | 7.87 | 23.87 | 38.95 | 18.68 | - | 10.30 | 16.10 | 25.50 | 15.30 | 2285 | 24.00 | 37.00 | 51.00 | 33.00 | - |
| ConvB [21] ♣ | 3.75 | 8.74 | 15.58 | 7.96 | 2792 | - | - | - | - | - | - | - | - | - | - |
| GCN [19] | 21.25 | 33.04 | 47.50 | 29.80 | - | 1.94 | 4.14 | 4.63 | 2.20 | 6450 | 10.00 | 18.10 | 30.00 | 16.40 | 600 |
| GCN+sim [19] | 21.33 | 33.46 | 46.75 | 30.03 | - | 0.01 | 0.13 | 0.20 | 0.13 | 6479 | - | - | - | - | - |
| *LM-augmented* | | | | | | | | | | | | | | | |
| GCN+BERT$_{large}$ [19] | 38.79 | 56.46 | 72.96 | 50.38 | - | - | - | - | - | - | - | - | - | - | - |
| GCN+BERT$_{large}$+sim [19] | 39.42 | 59.58 | 73.59 | 51.11 | - | - | - | - | - | - | - | - | - | - | - |
| BERT+ConvE [17] ◇ | 33.20 | 52.10 | 69.10 | 45.30 | 260 | 12.80 | 20.00 | 31.50 | 19.00 | 408 | 22.40 | 33.00 | 46.50 | 30.50 | 190 |
| BERT+ConvTransE [17] ◇ | 34.00 | 52.00 | 67.50 | 45.80 | 276 | 12.70 | 19.90 | 31.00 | 18.80 | **390** | 21.80 | 32.10 | 44.90 | 29.60 | 211 |
| BERT+DeepConv [17] ◇ | 41.80 | 61.00 | 77.20 | 54.00 | 161 | 12.70 | 19.70 | 31.40 | 18.80 | 422 | 24.60 | 35.40 | 48.80 | 32.70 | 190 |
| BERT-ResNet+RE [17] ◇ | 43.80 | 62.30 | 76.90 | 55.50 | 169 | 13.70 | 21.00 | 31.70 | 19.90 | 413 | 27.00 | 38.70 | 51.40 | 35.40 | 186 |
| BERT-ResNet+KD+RE [17] ◇ | 45.20 | 64.70 | 76.90 | 56.90 | 169 | 12.80 | 20.10 | 31.70 | 19.10 | 413 | 26.90 | 38.60 | 51.40 | 35.30 | 186 |
| RGAT+BERT$_{large}$ [13] ♣ | 27.90 | 47.74 | 67.21 | 41.03 | 177 | - | - | - | - | - | - | - | - | - | - |
| RGAT+BERT$_{large}$+sim [13] ♣ | 30.75 | 51.54 | 69.34 | 43.97 | 169 | - | - | - | - | - | - | - | - | - | - |
| StAR [36] | - | - | - | - | - | - | - | - | - | - | 26.60 | 40.40 | 56.20 | 36.50 | **117** |
| CNPC-S [41] | 45.33 | 61.46 | 75.92 | 54.52 | - | - | - | - | - | - | - | - | - | - | - |
| CNPC-I [41] | 48.29 | 65.04 | 79.13 | 59.00 | - | - | - | - | - | - | - | - | - | - | - |
| CoRPe [25] | 46.75 | 65.66 | 77.67 | 57.24 | 98 | - | - | - | - | - | 31.66 | 45.11 | 59.29 | 40.99 | 160 |
| CSProm-KG+MPIKGC-S [42] | - | - | - | - | - | - | - | - | - | - | 26.71 | 39.52 | 54.30 | 35.95 | 179 |
| LoLLM | 40.67 | 58.75 | 72.00 | 51.85 | 187 | 17.08 | 24.41 | 34.42 | 22.67 | 989 | 31.57 | 44.23 | 56.21 | 40.20 | 221 |
| B-LoLLM | **49.58** | **66.41** | **79.33** | **59.42** | **82** | **17.49** | **26.08** | **37.51** | **24.11** | 1358 | **32.16** | **45.64** | **59.62** | **41.37** | 214 |

in which $i$ is an entity in the KG. Eventually, to derive the score of the target triple $(s, r, o)$, we employ the decoding approach connecting entity and relation embeddings, which is illustrated as following:

$$score(s, r, o) = \phi_2(\mathbf{M}(\boldsymbol{\psi}_s, \boldsymbol{r})\mathbf{W}_{conv})\boldsymbol{\psi}_o, \quad (19)$$

in which $\mathbf{M}$ is the transformation matrix for all the kernals from the convolution [29]. $\boldsymbol{r}$ refers to the embedding of relation $r$, and $\mathbf{W}_{conv}$ is a linear transformation matrix. $\phi_2$ is the activation function, which is $\text{Sigmoid}(\cdot)$ in this method. As for the optimization of the reasoning process, we leverage the binary cross-entropy loss.

# 5 EXPERIMENTAL RESULTS
## 5.1 Datasets and Baselines
**Datasets.** We need incomplete KGs to illustrate the effectiveness of our proposed method. Especially, we desire some sparse KGs to evidently show the performance. Therefore, we leverage the sparse KG CN-100K, which is extracted from the original ConceptNet [30]. Moreover, we choose the FB15K-237-Sparse [17], which decrease the density of original FB15K-237 to simulate the structural sparsity. We also implement our method on FB15K-237 to illustrate the effectiveness on general KG. The detailed statistics are in Table 1.

**Baselines.** For comprehensive comparison, we select typical methods in KG embedding and reasoning, including structure-based DisMult [44], ConvE [5], ConvTransE [29], ConvKB [21], StAR [36]. Moreover, we choose the GCN [19] as well, including a GCN method with the densification strategy named GCN+sim. LM-augmented KG reasoning methods

contain GCN+BERT+sim [19], BERT+DeepConv, BERT+ConvE, BERT+ConvTransE, BERT+ResNet+KD+RE [17], RGAT+BERT and RGAT+BERT+sim [13]. Furthermore, we compare LoLLM with the most recent baselines CNPC-S, CNPC-I and CoRPe [25], and an LLM-based method CSProm-KG+MPIKGC-S [42] as well.

## 5.2 Metrics and Experimental Details
**Metrics.** We choose the ranking metrics in link prediction for multiple runs considering the random seeds and samples. We use Hits@$t$ to evaluate the target triple $(s, r, o)$ among negative samples and see if $(s, r, o)$ can rank the top 1, 3 and 10. Besides, we also use the mean reciprocal ranking (MRR) and mean rank (MR) to evaluate the overall ranking situation.

**Experimental Details.** For the reasoning paths, we preset the max length $L_m = 3$ considering the complexity. In the structural embedding process, we use a 2-layer GCN and set the dimension as 200, dropout as 0.3. As for the logical tuning embedding process, we utilize the GPT-3.5 [23] for supervision considering the convenience and cost. We set the batch size as 4 and use the Adam optimizer [15] with learning rate as $1.5e^{-5}$. In the LL function, we set $\eta = 4$ as the margin of LoLLM and $\eta = 16$ for B-LoLLM. As for hyperparameter $\lambda$ adjusting TL and LL in Eq. (16), we select $\lambda = 0.2, 0.7$ in CN-100K and FB15K-237-Sparse respectively. We implement the experiments on one NVIDIA's Tesla V100 graphic card. More details of the settings are in Appendix C.2.

## 5.3 Comparison Results
In Table 3, we use the ranking task to illustrate the effectiveness of LoLLM compared with other methods. LoLLM is mainly evaluated

**Table 4: Ablation Results on CN-100K and FB15K-237-Sparse.**

| Model | CN-100K | | | | FB15K-237-Sparse | | | |
|---|---|---|---|---|---|---|---|---|
| | H@1 | H@3 | H@10 | MRR | H@1 | H@3 | H@10 | MRR |
| B-LoLLM | **49.58** | **65.58** | **79.33** | **59.42** | **17.49** | **26.08** | **37.51** | **24.11** |
| -w/o LTE | $18.83_{\downarrow 30.75}$ | $32.58_{\downarrow 30.00}$ | $44.83_{\downarrow 34.50}$ | $27.39_{\downarrow 32.03}$ | $0.01_{\downarrow 17.48}$ | $0.05_{\downarrow 26.03}$ | $1.19_{\downarrow 36.32}$ | $0.51_{\downarrow 23.60}$ |
| -w/o SE | $32.67_{\downarrow 16.91}$ | $49.42_{\downarrow 16.16}$ | $68.00_{\downarrow 11.33}$ | $43.49_{\downarrow 15.93}$ | $17.06_{\downarrow 0.43}$ | $24.17_{\downarrow 1.91}$ | $33.43_{\downarrow 4.08}$ | $22.71_{\downarrow 1.40}$ |
| -w/o LL | $44.50_{\downarrow 5.08}$ | $62.58_{\downarrow 3.00}$ | $76.50_{\downarrow 2.83}$ | $55.70_{\downarrow 3.72}$ | $16.13_{\downarrow 1.36}$ | $21.87_{\downarrow 4.21}$ | $30.65_{\downarrow 6.86}$ | $20.87_{\downarrow 3.24}$ |
| -w/o LL & TL | $40.33_{\downarrow 9.25}$ | $59.00_{\downarrow 6.58}$ | $73.25_{\downarrow 6.08}$ | $51.63_{\downarrow 7.79}$ | $14.51_{\downarrow 2.98}$ | $18.32_{\downarrow 7.76}$ | $23.26_{\downarrow 14.25}$ | $17.75_{\downarrow 6.36}$ |

**Table 5: Comparison Results (%) of KG Reasoning on CN-100K with LLMs.**

| Model | Embedding | LLM-based | H@1 | H@3 | H@10 |
|---|---|---|---|---|---|
| CoRPe | ✓ | | 46.75 | 65.66 | 77.67 |
| CNPC-I | ✓ | | 48.29 | 65.04 | 79.13 |
| GPT-3.5 | | ✓ | 17.67 | 24.67 | 30.67 |
| GPT-4 | | ✓ | 28.50 | 37.92 | 42.92 |
| B-LoLLM | ✓ | ✓ | **49.58** | **66.41** | **79.33** |

on three datasets. The results of baselines without illustration are from published papers on top conferences and journals.

*5.3.1 Comparison with Embedding Methods.* Compared to all the **structure-based** methods, LoLLM achieves obvious improvement on all the datasets in Table 3. For the situation lacking structural information, the methods merely based on structure of KGs have difficulties in embedding entities and relations. Specifically compared to ConvE [29], which derives better performance, introducing logical information by reasoning paths in LoLLM obtains 32.87% average boost on CN-100K. On FB15K237-Sparse, LoLLM can still obtain as much as 7.52% average improvement of the reasoning performance. It illustrates that logical semantics can significantly reduce the influence of structural information deficiency.

As for the **LM-augmented** methods, they generally derive better results on solving KG reasoning lacking structural information. From Table 3, LoLLM and B-LoLLM can also own optimal results in KG reasoning task on three datasets. In particular, our methods obtain 13 out of 15 better KG reasoning results than LM-augmented methods. B-LoLLM results in as much as 1.48% average boost compared to CoRPe on CN-100K, which is the most recent baseline. Moreover, CNPC-S, CNPC-I and CoRPe all implement densification in relieving the influence of structure deficiency, which will consume much over time than LoLLM.

Overall, these results indicate the effectiveness and development in KG reasoning of our method, which introduces logical semantics by reasoning paths into the KG embedding process.

*5.3.2 Comparison with LLMs.* We also implement the KG reasoning on LLMs for comparison. As shown in Table 5, we compare the results of B-LoLLM with those of GPT-3.5 and GPT-4 [23]. CoRPe and CNPC-I are the most recent embedding-based methods. It is illustrated that the method fusing embedding module and LLM can obtain appearently better Hits@1, Hits@3 and Hits@10 results. To assess the KG reasoning capability of the LLM, we construct a zero-shot question answering scenario for the CN-100K test set and evaluate whether the LLM could rank the answers within top 1, 3, and 10. The results are also shown in Table 5. LLMs can not obtain

the performance as embedding-based methods. The phenomenon might be caused by the semantic gap between the discrete KGs and the continuous decoder-based LLMs.

## 5.4 Ablation Studies

We investigate the impacts of logical embedding tuning, TL and LL loss in the tuning process. As shown in Table 4, we rerun the models without the factors respectively, which are denoted as: **(1) w/o LTE** refers to the method removing logical tuning embedding (LTE) module in LoLLM. **(2) w/o SE** indicates the method removing structural embedding (SE) module in LoLLM. **(3) w/o LL** indicates the method removing logical loss (LL) during logical tuning embedding process. **(4) w/o LL & TL** indicates the method removing textual loss (TL) compared to **w/o LL** during logical tuning embedding process, and use the masked language modeling loss for fine-tuning instead.

The ablation results in Table 4 illustrates the effectiveness of important factors in our proposed method. **(1) The connecting of logical tuning embeddings and structural embeddings makes the method obtain better performance.** According to the the results, removing LTE and SE will all reduce the reasoning results of two datasets. **(2) The logical tuning embedding module is more effective than the structural embedding module.** As shown in Table 4, the results of B-LoLLM w/o LTE show a more obvious reduction compared to B-LoLLM w/o SE, which indicates that our proposed LTE is more critical in sparse KG reasoning, especially on FB15K-237-Sparse. **(3) The method gets optimal results when all the factors work simultaneously.** The reduction between B-LoLLM and B-LoLLM w/o LL & TL indicates their simultaneous influence for KG reasoning. **(4) TL and LL are all important for KG reasoning.** The results of B-LoLLM and B-LoLLM w/o LL indicates the importance of the logical semantics. The results of B-LoLLM w/o LL and B-LoLLM w/o LL & TL shows the importance of the textual semantics.

## 5.5 Weight Analysis

*5.5.1 Analysis of Language Models.* During the obtaining of logical tuning embedding, the choice of PLM is important for the reasoning results. We implement the LoLLM on T5 [27] and BERT [6], which have the encoding part in the model. The KG reasoning results on CN-100K are shown in Fig. 5 (a). It indicates that the reasoning constructed by the logical tuning embedding on BERT obtains much better results than that on T5. It may be because T5 owns strong capability on downstream tasks, which is not as good at providing and fine-tuning embedding as the pure encoding model. Therefore, we choose BERT as the PLM in implementing our method.

| Query | (Attend school, usedFor, Learn) ? | | |
|---|---|---|---|
| Reasoning Paths | Attend school has a prerequisite to teacher, teacher desires teach, teach causes learn.
Attend school causes learn.
Attend school causes education, education will locate at university, university will be used for learn.
Attend school causes education, education will be used for learn. | | |

| First-order Logic | LoLLM | w/o LLM | First-Order Logic |
|---|---|---|---|
| | **0.272** | 0.999 | hasPrerequisite$(X, Z_1)$ ∧ Desires$(Z_1, Z_2)$ ∧ Causes$(Z_2, Y)$→ usedFor$(X, Y)$ |
| | **0.234** | <0.001 | Causes$(X, Y)$→ usedFor$(X, Y)$ |
| | **0.234** | <0.001 | Causes$(X, Z_1)$ ∧ locatedAt$(Z_1, Z_2)$ ∧ usedFor$(Z_2, Y)$→ usedFor$(X, Y)$ |
| | **0.260** | <0.001 | Causes$(X, Z)$ ∧ usedFor$(Z, Y)$→ usedFor$(X, Y)$ |

**Figure 4: A case indicates the effectiveness of LLM during reasoning.**

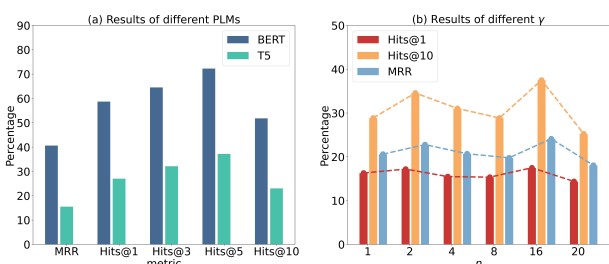

**Figure 5: KG reasoning results based on different parameters. (a) represents the impact of encoding model; (b) represents the impact of margin $\gamma$.**

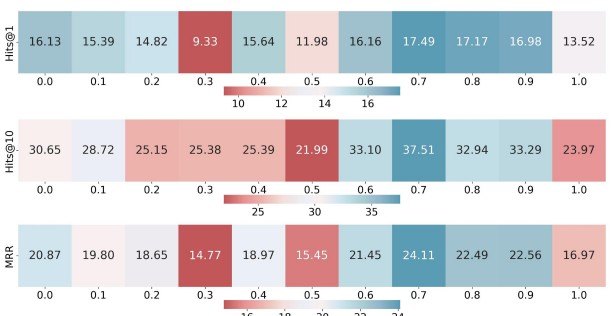

**Figure 6: KG reasoning results (%) based on different hyper-parameters $\lambda$.**

*5.5.2 Analysis of margin $\gamma$.* The margin $\gamma$ is a parameter in LL from Eq. (15) during logical tuning embedding. The margin determines the distance between positive and negative samples in the optimization. As shown in Fig. 5 (b), we record the KG reasoning results with $\gamma = \{1, 2, 4, 8, 16, 20\}$ on FB15K-237-Sparse. From the results, it is figured that $\gamma$ can influence the performance of reasoning. When $\gamma = 16$, all the metrics obtain best results on FB15K-237-Sparse. This phenomenon indicates that a margin that is either too large or too small will result in a negative effect.

*5.5.3 Analysis of hyper-parameter $\lambda$.* In LoLLM, $\lambda$ performs a significant role in adjusting LL and TL in logical tuning embedding process from Eq. (16). We rerun the method on FB15K-237-Sparse and record the KG reasoning results for different values of $\lambda \in [0.0, 1.0]$ with a step of 0.1. The result distributions are shown in Fig. 6. From the results, we can figure that the reasoning performance varies with different values of $\lambda$ and the distributions of all the metrics are generally consistent. In particular, the performance when $\lambda \in [0.6, 0.9]$ is better than that when $\lambda \in [0.3, 0.5]$. When $\lambda = 0.7$, the performance of our method gets the peak in KG reasoning. Based on these results, we choose $\lambda = 0.7$ during the logical tuning embedding process.

## 5.6 Case Study

As we discussed in Section 1 and Section 4, the logic rules and their confidences will help the method improve the embeddings and reasoning results. When introducing the logics, the confidences are considered as the reliability. We record the confidences generated by the LLM and the attention weights respectively, which is widely used in previous rule learning methods [38]. The results are shown

in Fig. 4. In predicting the query $(Attend school, usedFor, ?)$, the four first-logic rules impact differently in the reasoning process. Without the supervised information (named as w/o LLM), we calculate the attention weights following the attention weight of previous rule learning methods, but the confidence of hasPrerequisite$(X, Z_1)$ ∧ Desires$(Z_1, Z_2)$ ∧ Causes$(Z_2, Y)$ → usedFor$(X, Y)$ is up to 0.999, which is not consistent as the reality. Therefore, the confidence of logic according to embeddings is not reliable to be the supervised information. These results indicate the significance of LLM supervision.

## 6 CONCLUSION AND FUTURE WORK

In order to solve the sparse structure in KG reasoning, we propose a logic-aware method LoLLM with LLM supervision to handle the deficient supervision and semantic gap in implementation. LoLLM firstly obtains structural embeddings by the GCN with attention. Secondly, we construct reasoning paths based on the first-order logics, and inject the semantics through an LLM-enhance tuning process. In this process, we derive logical tuning embeddings of the KG. Finally, LoLLM connects structural embeddings and logical tuning embeddings and implements the prediction. Extensive experiments on two sparse KGs and a general KG show that LoLLM achieves outstanding effectiveness compared to SOTA structure-based and LM-augmented baselines.

LoLLM is still expected to be improved in more reasoning scenarios. Despite the sparse KGs, we would like to capture the challenges of more benchmark KGs and expand LoLLM to solve more general issues in KG reasoning.

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

## A MORE DETAILS OF B-LOLLM

Because of the limitation of pages, we illustrate the loss of B-LoLLM during logical tuning embedding process here. Following the similar process, we aggregate the hidden states of multiple $\langle s \rangle$ tokens in $\mathcal{T}(s, r, o)$ and leverage the mean pooling to derive the embedding $z_s$ of $s$, whose process is as following:

$$z_s = \sum_{k=1}^{|\langle s \rangle|} z_{\langle s \rangle, k}, \tag{20}$$

$$z_{\langle s \rangle, k} = \mathcal{M}(\langle s \rangle_k | \mathcal{T}(s, r, o)), \tag{21}$$

The confidence still needs to be in $[0, 1]$, so we derive the LLM-based embedding of $\langle s \rangle$:

$$w_{k,s} = \text{LLM}(\mathcal{T}(s, r, o)), \tag{22}$$

$$\overline{w}_{k,s} = \text{softmax}(w_{k,s}) = \frac{\exp(w_{k,s})}{\sum_{k'=1}^{|\langle s \rangle|} \exp(w_{k',s})}, \tag{23}$$

$$\overline{z}_s = \overline{w}_{k,s} \cdot z_{\langle s \rangle, k}, \tag{24}$$

In order to introduce the logical information, the LL of B-LoLLM is calculated by following equations:

$$
\begin{aligned}
\mathcal{L}_{LL} = {} & \theta_1 \max(0, D(z_s, \overline{z}_s) - D(z_{s^-}, \overline{z}_s) + \gamma) \\
& + \max(0, D(z_o, \overline{z}_o) - D(z_{o^-}, \overline{z}_o) + \gamma) \\
& - \theta_2 \log \frac{\exp(\overline{z}_s \cdot z_{s,m}/\mu)}{\sum_i \exp(z_{s,i^-} \cdot z_{s,m}/\mu) + \exp(\overline{z}_s \cdot z_{s,m}/\mu)} \\
& - \log \frac{\exp(\overline{z}_o \cdot z_{o,m}/\mu)}{\sum_i \exp(z_{o,i^-} \cdot z_{o,m}/\mu) + \exp(\overline{z}_o \cdot z_{o,m}/\mu)},
\end{aligned}
\tag{25}
$$

---

**Algorithm 1** Process of KG reasoning by LoLLM.

**Input:** KG $G \langle E, R, T \rangle$ and target triple $(s, r, o)$, Max length of relational paths $L_m$, hyper-parameters $\lambda, \gamma$, templates $\tau(\cdot)$, etc.

**Output:** Score of $(s, r, o)$ and first-order logic rules with confidences $w$.

1: **for** each triple $(s, r, o)$ **do**
2:   $\mathcal{P}_{s,o} \leftarrow$ extract reasoning paths within the length $L_m$ from $s$ to $o$.
3:   $\mathcal{T}(s, r, o) \leftarrow$ construct the reasoning paragraph by the template $\tau(\cdot)$ by Eq. (6), (7).
4:   $z_o \leftarrow$ mean embedding of the $\langle o \rangle$ in $\mathcal{T}(s, r, o)$ by Eq. (8), (9).
5:   $\overline{w}_k \leftarrow$ derive confidence values of reasoning paths from LLM by Eq. (11).
6:   $\mathcal{L}_{TL} \leftarrow$ calculate textual loss (TL) of $\mathcal{T}(s, r, o)$ by Eq. (13).
7:   $\mathcal{L}_{LL} \leftarrow$ calculate logical loss (LL) of $\mathcal{T}(s, r, o)$ by Eq. (15).
8:   $\mathcal{L} \leftarrow$ weighted summation of $\mathcal{L}_{TL}, \mathcal{L}_{LL}$ by $\lambda$ following Eq. (16).
9:   Update the parameters by Adam optimizer.
10:   $h_{\langle o \rangle} \leftarrow$ obtain logical tuning embeddings of the object $o$.
11: **end for**
12: **for** each triple $(s, r, o)$ **do**
13:   **for** $l$ in $L$ layers of GCN **do**
14:     $v_i^{(l+1)} \leftarrow$ message passing from $v_j^{(l)}$ by Eq. (4), in which $j \in \mathcal{N}_i^r$.
15:   **end for**
16:   Concatenate structural embedding and logical tuning embedding as $\psi$.
17:   Calculate score of $(s, r, o)$ by Eq. ((19)).
18:   Update parameters by binary cross entropy loss.
19: **end for**
20: **return** Scores of target triples and first-order rules with confidence $\overline{w}_k$.

---

in which $\theta_1, \theta_2 = \{0, 1\}$. The value is determined by the experimental results. The overall loss is still designed as:

$$\mathcal{L} = \lambda \mathcal{L}_{LL} + (1 - \lambda) \mathcal{L}_{TL}. \tag{26}$$

## B TRAINING PROCESS

LoLLM can predict missing entities of incomplete triples, and also generate confidences of the logics during reasoning. In Algorithm 1, we demonstrate the process of reasoning by LoLLM. LoLLM uses a KG as the input and at last outputs the score of target triple and a set of first-order rules with confidences. The detailed reasoning process is in Algorithm 1.

## C EXPERIMENTAL DETAILS

### C.1 Baselines

For comprehensive comparison, we select typical models in KG embedding and reasoning methods. We divide these methods into structure-based and language model (LM)-augmented. Structure-based methods mainly use the structural information to embed the entities and relations for the reasoning. The methods include a typical method DisMult [44] based on semantic factorization and a convolution-based method ConvE [5]. ConvTransE [29] is a method

```
-- Background: We are going to evaluate the corresponding score between the exact entity in the
context and the same entity in the sentence.

-- Input:
{
    "target entity": [The entity to be evaluated],
    "target sentence": [The standard sentence],
    "reasoning paragraph": [Multiple sentences about "target entity"],
    "number": [Numbers of "target entity" in the "reasoning paragraph".]
}
(Clarification: The inputs are json objects. The "number" of "target entity" in the reasoning
paragraph means the complete token. E.g. "book" in the "bookstore" does not count.)

-- Output:
{
    "scores": [A list containing values of scores. The scores evaluating the similarities of the
meanings between "target entity" in the "target sentence" and "reasoning paragraph". The scores
are in float type. The sum of all values in scores is 1.]
}
(Clarification: The outputs are json objects. The length of "scores" is strictly equal to the
"number" in the input.)

-- Examples:
    \INPUT:
    {
        "target entity": "shop",
        "target sentence": "Shop is used to buy present.",
        "reasoning paragraph": "Shop is used to buy present. There is a shop. I want to shop.",
        "number": 3
    }

Your Answer:
    \OUTPUT:
    {
        "scores": [0.5, 0.1875, 0.3125]
    }

-- Clarifications:
    1. Please check the inputs. The inputs in wrong formats are handled as error cases.
    2. The input and output are all in format of json.
    3. We do not need the reason. Please only output the scores.
    4. Please increase the variance of the scores. E.g. [0.6, 0.4] is better than [0.5, 0.5] when
the "number" is 2.
    5. The length of "scores" is strictly equal to the "number" in the input.

-- Task:
    Please strictly follow the above content and requirements to answer accordingly based on my
inputs, and answer only according to the output and reject any other descriptive text.

-- Confirmation:
    If you understand the above requirements, please answer "yes" only.
```

**Figure 7: Prompt for generation of $w_k$.**

based on ConvE and the translational TransE. ConvKB [21] leverages convolutional neural networks (CNNs) to capture the global relations and transitional properties. Moreover, we choose the GCN [19] as well, including a GCN method with the densification strategy named GCN+sim.

LM-augmented methods introduce pretrained language models to enhance the results of KG reasoning. Typically, GCN can be connected with the embeddings derived from the PLM for the KG reasoning [19], which is named as GCN+BERT+sim. The model utilizes a student reranking network to develop a deep convolutional baseline named BERT+DeepConv. It is also improved with the ResNet, ranking ensemble and knowledge distillation,

named BERT+ResNet+KD+RE [17]. The models RGAT+BERT and RGAT+BERT+sim [13] are based on a relational graph attention network and a PLM. Furthermore, we add the most recent baselines CNPC-S, CNPC-I and CoRPe [25] as well.

## C.2 Experimental Settings

Since various parameters can affect the performance of LoLLM on different datasets, we tune the parameters separately for each dataset during the logical tuning embedding module. Our search space for parameters is as follows:

- Learning rate: {0.00001, 0.000015, 0.000002}

**Table 6: Comparison sesults of KG reasoning with different prompts on FB15K-237-Sparse.**

| Model | H@1 | H@3 | H@10 | MRR | MR |
|---|---|---|---|---|---|
| LoLLM-Scale(0, 4) | 16.96 | 24.07 | 32.70 | 22.37 | 1414 |
| LoLLM-Weights | **17.08** | **24.41** | **34.42** | **22.67** | **989** |

- Margin hyper-parameter $\gamma$ in LL: {1, 2, 4, 8, 16, 20}
- Number of negative samples in LL: {1, 2, 3, 4}
- Temperature of infoNCE loss in LL: {1, 2, 3}
- Batch size: {2, 4, 8}

## C.3 Prompt for generation of $w_k$

Fig. 7 shows the prompt we use for the generation of $w_k$ during LLM-enhanced tuning process. The input and output are all in json format. According to this prompt, we obtain the scores of similarity between the target sentence and reasoning paragraph. The prompt works as follows:

- We firstly summarize the background of the prompt.
- Input: The sentence $\tau(s, r, o)$ and reasoning paragraph $\mathcal{T}(s, r, o)$ with the target triple $(s, r, o)$ are as the input.

Moreover, we provide the number of scores in the output to avoid the hallucination of the LLM [11].

- Output: The LLM outputs a list of scores evaluating the similarity between $o$ in $\tau(s, r, o)$ and $\mathcal{T}(s, r, o)$.

In particular, to reduce the impact of a single reasoning sentence in $\mathcal{T}(s, r, o)$, we concatenate $\tau(s, r, o)$ with $\mathcal{T}(s, r, o)$ when evaluating and aggregating. Considering the possible hallucination, we specially declare to increase the variance of the scores. E.g. "[0.6, 0.4] is better than [0.5, 0.5]". Moreover, according to the strategy that leverages LLM to evaluate the scale [50], we also implement the prompt that generates a score in a scale of [0, 1, 2, 3, 4]. The results are in Table 6. It is shown that the reasoning results of LoLLM-Weights is slightly better than those of LoLLM-Scale(0, 4). The reason might be that LoLLM-Scale(0, 4) have finite values for evaluating, which is not enough for better distinguishing.

## D CODE APPENDIX

For reproducibility, core codes of LoLLM are in an anonymous hyperlink: LoLLM. We will make all the available source codes open for the method upon publication.

Received 20 February 2007; revised 12 March 2009; accepted 5 June 2009

