# OpenReview forum: "Logic-Aware Knowledge Graph Reasoning for Structural Sparsity under Large Language Model Supervision"
_ACM.org/TheWebConf/2025/Conference — WWW 2025 Poster_

### Official Review · Reviewer_HRJz · 2024-11-13

**Novelty:** 4
**Technical Quality:** 4

**Review:**

This paper introduces a novel KG reasoning approach LoLLM, which consists of two modules, 1) Logical Tuning Embedding and 2) Structural Embedding, to inject logic with information from LLMs.

Pros:
* The motivation is well discussed.
* The overall writing is good.

Cons:
* The overall idea is a little confusing. The novel losses of TL and LL heavily rely on the generation of similarities from the LLM, and the authors did not mention the exact LLM they used in the main experiments.
* The performance improvement is not significant; it would be better to include a statistical significance analysis.
* In the ablation study, I think B-LoLLM without LTE is identical to GCN [19], but B-LoLLM without LTE (in Table 4) underperforms GCN (in Table 3). Any explanation for this finding?

**Questions:**

1. In Line 448, in the set of reasoning paths $P_{(s,o)}$, each reasoning path should start with $s$, not $x$, and end with $o$, not $y$.
2. Using the BFS algorithm to iterate KG for reasoning paths could introduce noise for KG reasoning. Can you provide some discussion on that?
3. What is the inference cost of querying LLM for generating similarities in Fig 3, if you choose ChatGPT or GPT-4 as the LLM backbone?

**Reviewer Confidence:**

3: The reviewer is confident but not certain that the evaluation is correct

**Scope:**

4: The work is relevant to the Web and to the track, and is of broad interest to the community

---

### Official Review · Reviewer_pZ2x · 2024-11-26

**Novelty:** 5
**Technical Quality:** 4

**Review:**

The paper proposes a new KG inference method that utilizes supervised information injection logic provided by large models.
### pros:
1. The paper is well written with clear logic
2. The innovation of the paper is good
### cons:
1. The comparative experiments in the paper were not sufficiently conducted
2. Each component of the article requires corresponding ablation experiments and analysis

**Questions:**

1. Why use GCN instead of other models when extracting structural information? Is it necessary to analyze the selection of structural information extractor
2. When presenting the results, WN18RR, as a traditional binary knowledge graph, should also be included as part of the experiment
3. In the ablation experiment, the ablation results of FB15K-237 also need to be presented

**Reviewer Confidence:**

3: The reviewer is confident but not certain that the evaluation is correct

**Scope:**

3: The work is somewhat relevant to the Web and to the track, and is of narrow interest to a sub-community

---

### Official Review · Reviewer_C7Zh · 2024-11-26

**Novelty:** 5
**Technical Quality:** 4

**Review:**

This paper proposes a KG reasoning method that integrates large language models (GPT), small language models (BERT), and GNN, effectively solving the problem of the current KG reasoning model being difficult to obtain meaningful reasoning path context on sparse KG, significantly improving the model's reasoning performance on sparse KG.

Pros:
1. Clear motivation analysis.
2. Sufficient experiments to validate the effectiveness of the method.
3. The idea is relatively novel, using a large model to guide the KG inference framework of GNN+BERT.

Cons:
1. Some method settings may be questionable.
2. Lack of theoretical analysis on the effectiveness and reliability of the GPT scoring framework.
3. The description of the method section is not comprehensive enough.

**Questions:**

1. The purpose of this article is to predict entities in incomplete triples, but when sampling the inference paths of triples, the author provides clear head and tail entities for BFS (Line 443 in Page 4). **I don't quite understand how these inference paths are sampled during testing?** After all, we need to assume that the head entity or tail entity is unknown. Also, the template in Figure 3 clearly indicates the target entity, which seems impossible to achieve in the actual testing process?
2. The relationship embedding $r$ in Figure 2 is not explicitly given in the method section. Is it $W_r$ in Eq. 4? In addition, is $\theta_r$ in Eq. 5 a trainable or manually set parameter?
3. On page 4, line 455, the author states the use of LLM to generate a textual description of the corresponding reasoning path, but there is no prompt template provided in the main text or appendix.
4. The description of B-LoLLM is not clear, although according to the equations in Appendix A, we can guess that B-LoLLM adds prediction loss to the head entity based on LoLLM. However, as the most effective model, I suggest the author briefly summarize it in the main text to clearly distinguish its differences from LoLLM.
5. Although some current work is indeed based on GPT for atomic reasoning decision-making (such as ToG), most of it belongs to the field of training-free in-context learning. This paper uses the scores provided by GPT as one of the training criteria, which has a significant impact on the parameter optimization of the model. **Therefore, I am curious about how the author can ensure the rationality of GPT's scoring of inference paths, or can some theories be used to optimize the boundary analysis of the objective function injected with GPT scores?**

**Reviewer Confidence:**

3: The reviewer is confident but not certain that the evaluation is correct

**Scope:**

4: The work is relevant to the Web and to the track, and is of broad interest to the community

---

### Official Review · Reviewer_kJuB · 2024-11-30

**Novelty:** 4
**Technical Quality:** 6

**Review:**

This work proposes a novel solution for the task of link prediction on sparse KGs using supervision from large language models (LLMs). The work is motivated by the fact that existing solutions (e.g., embedding-based approaches, path- or rule-based methods) struggle at capturing the structure of sparse knowledge graphs. The approach, called LoLLM, provides not only answers to link prediction queries but can also provide explanations in the form of Horn rules. To do so, LoLLM first learns classical KGEs based on convolutional GNNs that are then fine-tuned using an LLM, guided by a set of reasoning paths that take the form of Horn rules. The experimental evaluation, conducted on three benchmark datasets, suggests that the proposed approach outperforms both classical KGEs and LM-augmented methods for link prediction. Further studies, including an ablation study and a study of the impact of the hyper-parameters, provide evidence of the synergistic behavior of the different components of LoLLM.

LoLLM builds upon different powerful approaches to come up with an even more powerful solution for KG inference. I find its logic extraction approach particularly interesting and intuitive, moreover, the ablation study shows its practical values.

I have nevertheless a serious issue with this work. First of all, I believe the notion of sparsity is not properly discussed even though it is the main motivation of this work. KGs are usually sparse if we considered the normalized edge ratio, that is true. Also it is well-known that the node degree and triple predicate count usually exhibit a power-law distribution in large KGs, which has motivated some work on capturing knowledge for those sparse regions of the KG [1]. So a key question is, what does LoLLM does different than the other LM-augmented approaches that makes it (marginally) more suitable for sparse KGs? Is LoLLM's performance gain happening on those sparse areas?  The paper in its current version does not provide an answer to this question and seems more like an assembling of different blocks that happen to work on 3 datasets. I would urge the authors to provide more high-level lessons that shed light on the success of the LoLLM strategy.

Minor issues
By the end of the introduction the triple {Liquid, isA, Water} should be {Water, isA, Liquid}

[1] https://aclanthology.org/2024.naacl-long.183.pdf

**Questions:**

- What does LoLLM does different than the other LM-augmented approaches that makes it (marginally) more suitable for sparse KGs?
- Is LoLLM's performance gain happening on particularly sparse areas of the KG?

**Reviewer Confidence:**

3: The reviewer is confident but not certain that the evaluation is correct

**Scope:**

4: The work is relevant to the Web and to the track, and is of broad interest to the community

---

### Official Review · Reviewer_GtfF · 2024-12-02

**Novelty:** 5
**Technical Quality:** 4

**Review:**

This paper proposes a novel _Knowledge Graph_ (KG) reasoning approach LoLLM for injecting logic and information supplied by _Large Language Models_ (LLMs) in structural embeddings based on graph convolutional networks.

## Relevance

I deem the contribution relevant to the conference. Specifically, the paper proposes an approach to tackle the problem of sparsity in KGs which are very important resources on the web.

## Novelty

To the best of my knowledge, integrating logical rules into structural embeddings by means of BERT-like _Language Models_ (LMs) as well as assessing the confidence in rules through LLMs is a novel contribution. In contrast, the approach to combine the embedding from the BERT-like LM and the structural embeddings is based on concatenating the two embeddings: a relatively straightforward approach that may not fully address the complexity of the problem.

## Clarity/Readability

The structure of the paper follows a logical flow, beginning with a comprehensive introduction that outlines the motivation and significance of the research, followed by a detailed review of related works, and progressing through the methodology and experimental results. Each section is clearly delineated, with appropriate headings and subheadings that guide the reader through the content.

However, I suggest some adjustments in writing. Specifically, the paper defines _KG Reasoning_ (KGR) as predicting missing entities in incomplete triples. In contrast, this definition aligns more closely with _Link Prediction_ (LP), which is a specific task within the broad class of _KG Completion_ (KGC); KGR is even more general as it also includes other tasks beyond LP, e.g., consistency checking. Furthermore, it often adopts the expressions “deficient supervision”, “supervised information”, “supervised samples”, ”LLM supervision” which I personally find unrelated with the idea of enhancing logical rules with confidence scores. In addition, it often refers to rules expressed in logic as “logics”, in contrast, logic is the formal language used to express formulas, sometimes also representing rules. Moreover, the expressions “reasoning path”, “reasoning sentence”, and “reasoning paragraph” are adopted without prior definition. Finally, the proposals for future research seem a bit too general.

## Reproducibility
The paper provides a link to a folder containing data, code, and the fine-tuned LMs. However, it does not provide a file specifying the required python packages.

## Quality

The paper demonstrates fair technical soundness. Firstly, I deem the theoretical background adequate (except for the writing issues pointed out before). Moreover, the integration strategy is fairly well described and supported by mathematical formalizations. Nevertheless, I personally find that the reasons for converting logical rules to text before feeding them to an LLM should be clarified as possibly the LLM is capable of interpreting the logical formalism. Moreover, clarifications on the meaning of the term similarity (which measure?) can be provided in the prompt.

Moreover, the solution is very complex as it requires both a BERT-like LM and a LLM; therefore, its practical applicability may be limited.

The paper is well positioned with respect to the state-of-the-art (SOTA) in structure based embedding methods as well as LM augmented ones. Nevertheless, the positioning with respect to the SOTA logic-based approaches may be much more comprehensive and detailed. Specifically, Section 2.2 mentions solely logic-based approaches for tasks on images, text, and graphs rather than KGs which are the target of this paper. For instance, AnyBURL [1] and AMIE [2] are very important contributions in this field, and the analogies and differences of the proposed approach with respect to them should be reported. For what concerns AnyBURL, it is capable of learning rules also featuring confidence scores. Hence, it may be the case that it is not worthwhile to introduce additional complexity by prompting LLMs. Therefore, I deem worthwhile to investigate if the rules learned by AnyBURL can be integrated with structural embeddings by adopting the proposed strategy and to compare such approach with the rules with confidence scores assigned by an LLM.

Finally, a solid experimental evaluation comparing LoLLM with structure based as well as LM-augmented embeddings demonstrates good performance.

## Overall evaluation

In this section, I summarize the comments made above in a list of pros and cons.

### Pros

* very relevant
* fairly novel
* outperforms other approaches

### Cons

* limited clarity
* lack of essential references and positioning in the SOTA overview
* lack of some (not essential) details in the methodology
* the solution is very complex, as it requires both a BERT-like LM and an LLM
* the adoption of Large Language Models for generating the confidence scores of logical rules should be compared with other approaches furthering the same goal
* MINOR: the proposals for future work should be a bit more detailed
* MINOR: the repository does not contain a file listing all the required python packages

[1] Meilicke, Christian, et al. "Anytime bottom-up rule learning for large-scale knowledge graph completion." The VLDB Journal 33.1 (2024): 131-161.

[2] Galárraga, Luis, et al. "Fast rule mining in ontological knowledge bases with AMIE+." The VLDB Journal 24.6 (2015): 707-730.

**Questions:**

No questions

**Reviewer Confidence:**

3: The reviewer is confident but not certain that the evaluation is correct

**Scope:**

4: The work is relevant to the Web and to the track, and is of broad interest to the community